# Conflict-related intentional injuries in Baghdad, Iraq, 2003–2014: A modeling study and proposed method for calculating burden of injury in conflict

Guy W. Jensen[1]*, Riyadh Lafta[2], Gilbert Burnham[3], Amy Hagopian[4], Noah Simon[4], Abraham D. Flaxman[4]

1 University of California Davis Department of Surgery, Sacramento, California, United States of America, 2 Al Mustansiriya University, College of Medicine, Baghdad, Iraq, 3 The Johns Hopkins Bloomberg School of Public Health, Baltimore, Maryland, United States of America, 4 University of Washington School of Public Health, Seattle, Washington, United States of America

* jensengw@outlook.com

**Data Availability Statement:** All relevant data are within the manuscript and its Supporting Information files.

## Abstract

### Background

Previous research has focused on the mortality associated with armed conflict as the primary measure of the population health effects of war. However, mortality only demonstrates part of the burden placed on a population by conflict. Injuries and resultant disabilities also have long-term effects on a population and are not accounted for in estimates that focus solely on mortality. Our aim was to demonstrate a new method to describe the effects of both lives lost, and years of disability generated by a given conflict, with data from the US-led 2003 invasion and subsequent occupation of Iraq.

### Methods and findings

Our data come from interviews conducted in 2014 in 900 Baghdad households containing 5,148 persons. The average household size was 5.72 persons. The majority of the population (55.8%) were between the ages of 19 and 60. Household composition was evenly divided between males and females. Household sample collection was based on methodology previously designed for surveying households in war zones. Survey questions were answered by the head of household or senior adult present. The questions included year the injury occurred, the mechanism of injury, the body parts injured, whether injury resulted in disability and, if so, the length of disability.

We present this modeling study to offer an innovative methodology for measuring "years lived with disability" (YLDs) and "years of life lost" (YLLs) attributable to conflict-related intentional injuries, using the Global Burden of Disease (GBD) approach. YLDs were calculated with disability weights, and YLLs were calculated by comparing the age at death to the GBD standard life table to calculate remaining life expectancy. Calculations were also performed using Iraq-specific life expectancy for comparison.

**Funding:** This work was supported by Surgeons OverSeas, http://www.surgeonsoverseas.org/. RL received the funding. The funder had no role in study design, data collection and analysis, decision to publish, or preparation of the manuscript. The first author's work (GJ) on this project was supported in part by the National Center for Advancing Translational Sciences, National Institutes of Health, through grant number UL1 TR001860. The content is solely the responsibility of the authors and does not necessarily represent the official views of the NIH. The funder had no role in study design, data collection, decision to publish or preparation of the manuscript.

**Competing interests:** I have read the journal's policy and the authors of the this manuscript and have the following competing interests: AF has in the last 3 years consulted for Kaiser Permanente; Sanofi; Merck for Mothers; SwissRe; Agathos, Ltd (startup); and NORC (formerly National Opinion Research Council).

**Abbreviations:** DALY, disability-adjusted life year; DW, disability weight; GBD, Global Burden of Disease; PPES, probability proportional to estimated size; UI, uncertainty interval; YLD, year lived with disability; YLL, year of life lost.

We calculated a burden of injury of 5.6 million disability-adjusted life years (DALYs) lost due to conflict-related injuries in Baghdad from 2003 to 2014. The majority of DALYs lost were attributable to YLLs, rather than YLDs, 4.99 million YLLs lost (95% uncertainty interval (UI) 3.87 million to 6.13 million) versus 616,000 YLDs lost (95% UI 399,000 to 894,000). Cause-based analysis demonstrated that more DALYs were lost to due to gunshot wounds (57%) than any other cause.

Our study has several limitations. Recall bias regarding the reporting and attribution of injuries is possible. Second, we have no data past the time of the interview, so we assumed individuals with ongoing disability at the end of data collection would not recover, possibly counting more disability for injuries occurring later. Additionally, incomplete data could have led to misclassification of deaths, resulting in an underestimation of the total burden of injury.

## Conclusions

In this study, we propose a methodology to perform burden of disease calculations for conflict-related injuries (expressed in DALYs) in Baghdad from 2003 to 2014. We go beyond previous reports of simple mortality to assess long-term population health effects of conflict-related intentional injuries. Ongoing disability is, in cross section, a relatively small 10% of the total burden. Yet, this small proportion creates years of demands on the health system, persistent limitations in earning capacity, and continuing burdens of care provision on family members.

---

## Author summary

### Why was this study done?

- The Global Burden of Disease (GBD) project, based at the University of Washington, publishes the estimates of prevalence, incidence, years of life lost (YLLs), and years lived with disability (YLDs) for a comprehensive list of diseases and injuries and for all countries from 1990 to the present. The World Health Organization has also published YLL and YLD estimates.

- Previous researchers innovated the population-proportionate to size sampling methodology for household surveys in war zones, which we used in this study.

- Previous papers have attempted to calculate only Iraq War deaths, as mortality is an important population health indicator and is relatively straightforward to measure.

- Injuries are, along with mortality, an important factor in conflict settings; they measure a more nuanced toll on public health. Ongoing disability is important, especially when associated with demands on the health system, ongoing limitations in earning capacity, and the burdens created for family caregivers.

- We aim to lay out our application of the GBD methodology, along with our alteration to it for calculation of burden of injury in conflict. Additionally, we aim to describe specifically how these calculations were applied in hopes of developing a methodology for the study of conflict populations.

## What did the researchers do and find?

- This study innovates the application of GBD methodology for calculating YLLs and YLDs to data collected in a war zone.

- We calculated a burden of injury of 5.6 million DALYs lost due to conflict-related injuries in Baghdad from 2003 to 2014. These calculations encompass both mortality as well as the effects of injury and disability on the population of Baghdad. We performed these calculations using a variation of the GBD methodology to better reflect the dynamic nature of conflict.

- Media reports tend to focus heavily on the more "spectacular" aspects of conflict, such as explosions, shelling, and air strikes. By contrast, our results demonstrate gunshot wounds were the leading contributor to YLDs and YLLs.

## What do these findings mean?

- Our results demonstrate the importance of data-driven, epidemiologically sound assessment of conflicts, especially those spanning multiple years. However, it should be noted that our primary goal was to develop a methodology, demonstrated by our calculations with the city of Baghdad, to allow for its use in other conflicts.

- With more sophisticated methods, epidemiologists might better prevent the most harmful aspects of conflict for civilian populations, might better predict losses in advance of the onset of conflict, and might contribute to health systems in planning for care delivery.

- Limitations for this study include the possibility for recall bias among the survey participants as well as the possible undercounting of deaths due to injury due to classifying any injuries that did not definitively result in death as YLDs rather than YLLs. Lastly, we did not adjust for changes in the population of Baghdad over the course of the study.

## Introduction

Since the United States–led invasion of Iraq in 2003, the war has expanded to new territories and combatant groups. Attempts to track the health consequences of these events have grown increasingly sophisticated [1,2].

As conflicts in the region have escalated, Iraqi civilians have been displaced and have lost many previous gains in health and life expectancy [3–5]. Previous papers have attempted to calculate Iraq War deaths, as mortality is an important population health indicator and relatively straightforward to measure [6–9]. However, injuries and their resulting disabilities are an important factor in conflict settings as they measure a more nuanced effect on public health. Ongoing disability is important, especially when associated with health system demands, ongoing limitations in earning capacity, and the burdens created for family caregivers [10].

Previous papers have reported on interviews conducted during April and May 2014 in 900 households containing 5,148 persons. Elsewhere, we have described the 553 injuries reported by Baghdad residents in this sample population, 225 of which were conflict-related intentional injuries, and 328 were not directly related to conflict [11–14]. A third of the injuries reported in our study resulted in death, yet the fatality rate for conflict-related intentional injuries was higher (39%) than for unintentional injuries (7%). The major cause of injuries not related directly to conflict was falls (131), which increased dramatically since 2008, followed by traffic-related injuries (81). The proportion of injuries ending in disabilities remained stable through the 11-year survey period. This work focuses exclusively on conflict-related intentional injuries.

Our aim is to introduce a methodology for measuring disability-adjusted life years (DALYs) in a study of injuries and disabilities in a war setting using the Global Burden of Disease (GBD) methodology. For those who died, we calculated "years of life lost" (YLLs) based on both standard and Iraq-specific life expectancy. For those who survived, we calculated the burden of injury by applying disability weights (DWs) for varying types of injuries. Our intent is to encourage the field of war epidemiology to go beyond simple mortality counts and incorporate more sophisticated measures of injury and disability.

## Methods

### Data collection

The full methods and results from the initial data collection are previously published by Lafta and colleagues [11]. However, we have summarized our methods here because of the importance of the role of sampling on the ultimate results. We developed an Arabic language questionnaire to collect data on death, injury, and disability based on several previous injury or disability research efforts (S1 Questionnaire) [15–17]. Paper forms were used without identifying information. We asked respondents to name the year of the injury, limiting our scope to events between 2003 and 2014. Data were entered in SPSS (IBM, Armonk, New York), and files were rechecked against paper records to resolve discrepancies.

We defined injury as "any intentional or unintentional physical harm that required medical care, whether received or not, and with or without intervention, and which resulted in loss or reduction in normal activities for a period of time." Disability was defined as "any limitation to normal activities stemming from an injury." A conflict-related intentional injury was defined as being caused either by criminal activity or armed conflict. Criminal acts were defined as "events caused during robbery, assassinations, kidnapping, hijacking, or assaults likely carried out for personal reasons with the individual being specifically targeted or a bystander to a specifically targeted person." War or conflict-related events were defined as those occurring "in the course of wider indiscriminate intentional violence whether by militias or organized military and whether for sectarian or in the course of organized military operations." All injuries analyzed were war related according to the respondents.

One of our authors (RL) organized and trained 2 teams of Iraqi interviewers to administer questionnaires to selected households. The 8 interviewers (4 female, 4 male) and their 2 supervisors were community or family medicine doctors with field data collection experience. Interviews were conducted between May 8 and June 1, 2014.

The population-proportionate-to-size random sample included 30 clusters of 30 households in 14 administrative units in Baghdad, selected using a Google Earth mapping methodology described elsewhere [18].

## Sampling

Two-stage cluster sampling techniques were utilized to select 30 random clusters with a start household for each cluster. These clusters were created based on the sampling data for the 14 administrative units within the city of Baghdad. The clusters were created using a probability proportional to estimated size (PPES) technique. Beginning with the start household, teams used a systematic sample technique including every other house until 30 households were interviewed [11]. This sample size of the original survey was designed to measure an injury rate of 1% with a 95% confidence level [11].

## Burden of injury methodology

We applied modified GBD and DALY methodologies to this specific conflict-related data set. This work relied on the methods, values, and techniques described in the GBD 2013 study and regularly updated by the Institute for Health Metrics and Evaluation (IHME) [10]. DALYs were calculated by summing the YLLs and years lived with disability (YLDs) from participants due to injuries. The primary outcomes for this work were incidence YLLs and incidence YLDs.

YLDs were calculated by multiplying the duration of disability by DWs as defined for the GBD 2013 study [19]. YLLs were calculated by multiplying deaths by the remaining life expectancy at age of death based on a GBD standard life table for estimating premature mortality, rather than an Iraqi-specific life expectancy [20]. Because life expectancy represents the aspirational healthy life span for all individuals, we utilized the GBD standard life table created by selecting the lowest observed death rate for any age group in countries of more than 5 million in population [21]. This choice of a life expectancy standard is a values-based decision, in that it counts the "value" of all deaths worldwide the same regardless of nation or sex.

Because life expectancy differs by country and over time, we conducted a sensitivity analysis by estimating DALYs based on Iraqi life expectancy using World Bank estimates. In 2013, life expectancy at birth for Iraq was 69.4 years [22]. Methods did not otherwise differ from those used for calculations using the GBD life table.

## Analysis plan

The analysis was planned to utilize previously published GBD methods adapted to calculate incidental DALYs from an existing conflict data set. However, these calculations offered ample opportunity for error, and, therefore, to mitigate the possibility of error while applying our analysis, we selected at random a single patient with each injury pattern and performed the calculations by hand to ensure accurate calculations following complete analysis of the data (S1 Analysis Plan). Once these calculations were performed to determine the "correct" answer for each injury pattern, the methods for bootstrapping and variation of DWs with each repetition were developed later after complete data cleaning. However, as our work focuses primarily on the proposed methods for these calculations no data driven changes to the analysis occurred.

## Assessment of injury patterns and duration of disability

The calculation of DALYs from a specific data set requires aligning GBD methodology with data collection methods. YLDs require the use of differing DWs for differing injury patterns. The Baghdad injury data established 6 separate injury patterns, based on the key question ("Did the injured person suffer any disability that affects the ability to do normal activities for

some days?") and a follow-up question on the duration of disability (expressed in years, months, weeks, days, or continuing).

## Calculation of YLLs

For individuals who died between 2003 and 2014, we recorded age at time of death. This age was compared to the GBD life table to determine the difference between age at injury and life expectancy. YLLs were calculated when a person died of a conflict-related intentional injury, an outcome recorded on data collection. For 3 individuals, death was deemed either unrelated to the injury or only possibly related to the injury. These individuals were not included in the YLL calculation but were instead included in YLD calculations for the documented duration of their disability. YLLs have an outsized effect on total DALY calculations; therefore, conservative practice required YLLs to be calculated only for individuals who definitively died of their injury.

## Short-term injury

An injury was deemed to be "short term" when it persisted for less than 1 year [10]. Short-term injury DWs were used in the calculation of YLDs [23]. DALYs caused by short-term injury were calculated by multiplying the short-term injury DW for each patient by the duration of disability in fractions of years. Respondents reported the length of disability for short-term injury in months, weeks, or days. These values were divided by 12, 52, and 365, respectively, to create the fraction of the year used in the calculations.

## Long-term injury with recovery, both treated and untreated

Individuals who sustained injuries from which they recovered, but resulted in disability lasting a year or longer, formed the second injury pattern. We asked where injured individuals received care at the time of injury (hospital, health clinic, place of employment, or no care). For those who received no treatment, we calculated the burden of injury by multiplying the duration of the disability by the matched untreated DW. Individuals receiving any post-injury care were matched to the appropriate DW for treated patients; all levels of care were considered equal for this analysis.

## Long-term injury without recovery, both treated and untreated

Individuals who remained disabled at the time of data collection were classified as having permanent disability. We assumed these individuals would not have a shorter life expectancy but also would be unlikely to have a full recovery from their injuries. We also applied both treated and untreated DWs to these individuals. YLD calculations were made by multiplying treated and untreated DWs by the maximum life expectancy corresponding to the person's age at the time they were injured.

## Use of incidence versus prevalence DALYs

Readers familiar with the GBD methodology will note the calculations described here are based on the use of incidence, rather than prevalence YLDs. This represents the main deviation from the GBD methodology for application to a specific conflict data set. For the 2010 iteration of the GBD, the methodology was changed from using incidence-based YLDs to prevalence-based YLDs [10]. The reasoning for this change and the effects on the results are covered further in the discussion. YLLs remained incidence based, creating what some have referred to as

a hybrid approach [24]. We initially performed calculations with incidence YLDs; however, we also calculated prevalence YLDs for comparison.

## Bootstrapping

Burden of disease estimates have an inherent degree of uncertainty that arises from data collection, adjustments to the data, and statistical models. We would argue that the very nature of quantifying the burden of an individual's health state is also subject to some degree of uncertainty as an injury of both the same cause and nature may "burden" 2 individual patients to different degrees. Therefore, in accordance with GBD methods, uncertainty was propagated through the data set by repeating each calculation 1,000 times, with each drawing from the 1,000 draw distributions around each individual DW code.

Following data cleaning, 225 individuals remained who sustained a conflict-related intentional injury between 2003 and 2014. For purposes of matching DWs, we had sufficient information on 179 cases. To calculate YLD, we required either age at the time of injury for those injuries without recovery or the duration of disability from the injury for those injuries with recovery. Additionally, we needed to determine whether care was delivered. These 179 cases were resampled with replacement to create 1,000 separate samples. A randomly selected set of matched DWs from the appropriate DW table was mapped to each sample. We did not vary the DWs for cause-specific calculations, as variation in the DWs would not alter the final total. Summary statistics were then calculated from these 1,000 iterations. For cause-specific estimates, sampling with replacement was performed.

A total of 46 individuals with conflict-related intentional injury could not be used in the calculation of DALYs for the sample due to missing data points. For 29 of these individuals, the age at time of injury and duration of injury were missing. The remaining 17 individuals without death due to injury lacked the necessary information to pair the injury of the patient with a proper DW. The only injury mechanism overrepresented in those individuals detailed above was burn injuries. Otherwise, no injury mechanism was overly affected by the exclusions listed above.

## Extrapolation to the city of Baghdad

Following bootstrapping, we obtained the mean number of DALYs in the sample and divided this by the total number individuals that were sampled with replacement. We then multiplied the mean of the total DALYs by Baghdad population estimates. This was repeated for the lower and upper ends of the uncertainty interval (UI). Cause-specific DALY counts were generated in a similar fashion.

## Analysis

All analysis and data manipulation were performed using the R language for statistical computing [25]. We have provided the code and data set as a supplement (S1 Data Dictionary and R Code Publication and S1 DALY Data). This study is reported as per the Strengthening the Reporting of Observational Studies in Epidemiology (STROBE) guideline (S1 STROBE Guidelines Checklist) [26].

## Ethical approval

Ethical approval for the study was obtained from Mustansiriya University's scientific committee, and survey permission from the city of Baghdad. For security reasons, only verbal consent was obtained, with informed consent by participants recorded by the interviewers. Interviews

were conducted privately. The Institutional Review Boards of Johns Hopkins and the University of Washington determined the analysis of deidentified data as not human participant research.

## Results

Approximately 900 households were visited, and information was collected on 5,148 persons. This encompassed individuals who were living and those who had died from their injuries. The average household size was 5.72. The age breakdown of participants was as follows: 12% (*n* = 607) were under the age of 5, 44.2% (*n* = 2,274) were under 19, and 7.2% (*n* = 368) were over age 60. Household composition was evenly divided between males and females [11].

The initial sample contained 5,148 individuals, of whom 225 (4.4%) suffered conflict-related injuries between 2003 and 2014. Of these individuals, 88 died as a result of their injury, and 137 had injuries resulting in some degree of disability. Approximately 20% of these were excluded due to missing data required for the bootstrap analysis, leaving a final sample of 179 Baghdad household members sampled who were either injured or died due to conflict-related intentional injury.

We estimated 4,160 DALYs (95% UI 3,180 to 5,090) were lost due to conflict-related intentional injuries between 2003 and early 2014 among our surveyed respondents using standard life tables. Participants with conflict-related intentional injuries suffered an average 22 lost DALYs per injury (95% UI 19.5 to 26.9) or 0.81 (95% UI 0.51 to 1.12) per person. We estimated 3,700 YLLs overall (95% UI 2,800 to 4,500) or a mean 42 YLLs per death (95% UI 33 to 52). The mean number of YLLs for the sample was 0.72 per person (95% UI 0.54 to 0.87).

Among our respondents and their households, we estimated 450 YLDs (95% UI 296 to 663) resulting from conflict-related intentional injuries during the study period. The mean number of YLDs for the sample was 0.09 per person (95% UI 0.03 to 0.15) or 3.7 mean YLDs attributable to each injury (95% UI 2.4 to 5.3).

By multiplying our survey population rates by the total city population, we calculated a total municipal burden of conflict-related intentional injury. Because the initial survey clusters were assigned using PPES techniques, no post-survey weighting adjustments were required. The population of Baghdad at the time of data collection was estimated to be 6.90 million [27]. We estimate, therefore, 5.60 million DALYs (95% UI 4.30 million to 6.90 million) are owed to conflict-related intentional injuries between 2003 and early 2014 in the city of Baghdad. The burden of mortality in Baghdad from conflict-related intentional injury was 4.99 million YLLs (95% UI 3.87 million to 6.13 million). The estimated burden of injury for the city was 616,000 YLDs (95% UI 399,000 to 894,000) if calculated with incidence YLDs. The year-by-year estimates by DALYs, YLLs, and YLDs for the city of Baghdad are summarized in Fig 1.

For comparison, we performed the same calculation for conflict-related intentional injuries utilizing an Iraq-specific life expectancy. We estimate that 4,010 DALYs (95% UI 3,070 to 49,700) were lost from within the study sample. When extrapolated to the city of Baghdad as a whole, we estimate 5.41 million DALYs lost attributable to conflict-related intentional injuries (95% UI 4.14 million to 6.70 million).

As stated previously by our proposed method of calculation, we would estimate 450 incidence YLDs (95% UI 296 to 663) lost by the study population from 2003 to 2014. We repeated this calculation utilizing prevalence YLDs, counting them only for the time between the time of injury and the end of the study time frame. This resulted in a total of 49.7 prevalence YLDs (95% UI 31 to 72) lost over the course of the study. The year-by-year estimates of YLDs calculated by both incidence and prevalence techniques are summarized in Fig 2.

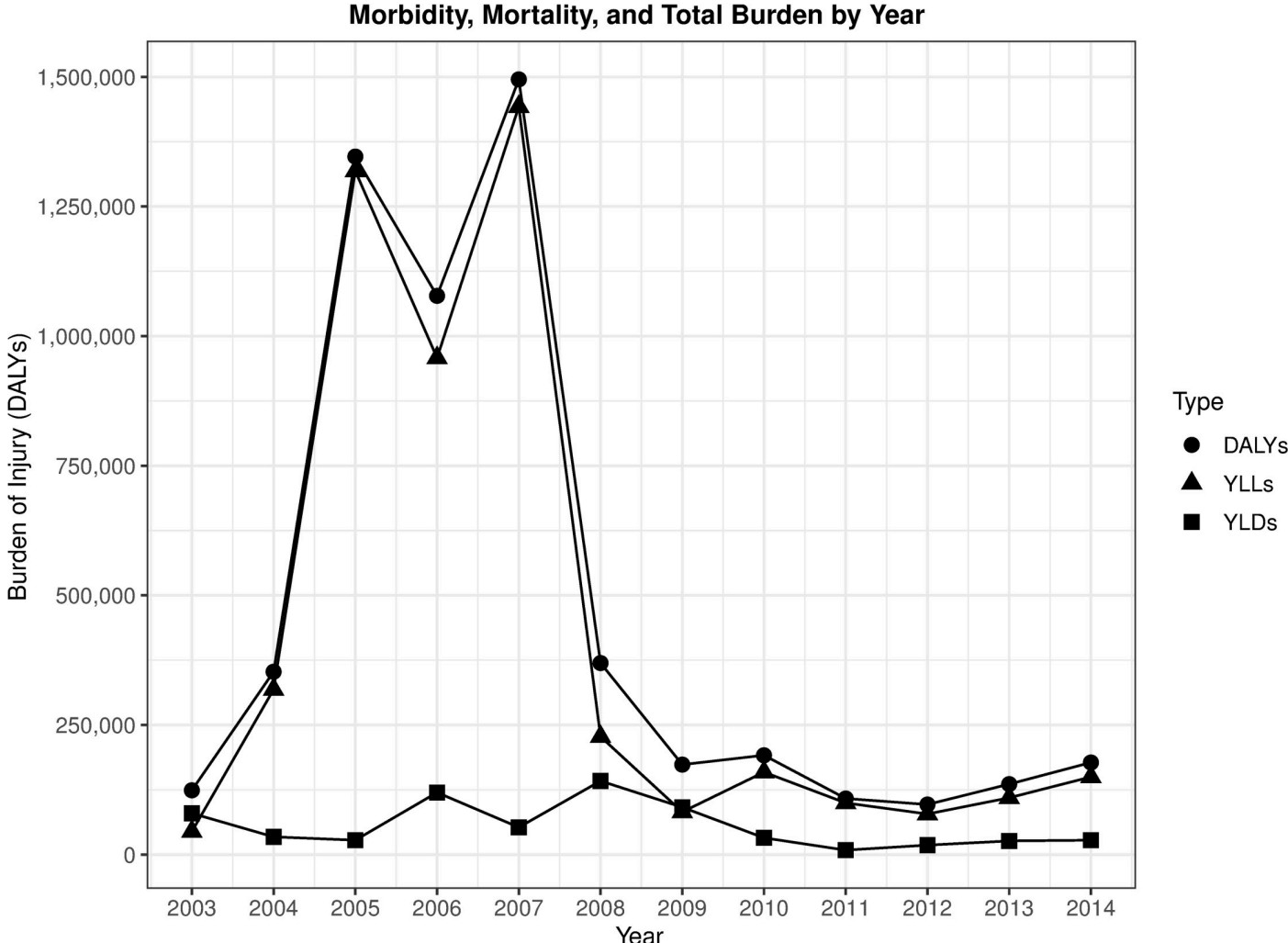

**Fig 1. Burden of conflict-related intentional injury by year in Baghdad, Iraq, 2003–2014, portraying total DALYs, as well as contributions of YLLs and YLDs.**
Source of data: survey of 900 Baghdad residents in May and June 2014, regarding household injury events between 2003 and 2014. DALY, disability-adjusted life year; YLD, year lived with disability; YLL, year of life lost.

Table 1 and Fig 3 portray the 7 causes of conflict-related intentional injury identified. Only one case of death by torture was named (accounting for 58 DALYs lost), although additional cases may have been missed because of a reluctance to report. Gunshot wounds made up the majority (57%) of all DALYs. Approximately 10% of the sample DALYs were attributable to blasts and explosions, with shelling and unspecified mechanisms accounting for 16% and 14%, respectively.

## Discussion

We estimate 5.6 million human DALYs were lost among residents of the city of Baghdad as a result of conflict-related intentional injuries over the course of the US-led coalition invasion of Iraq and subsequent insurgency. Our aim was to demonstrate a statistical method for describing the cumulative effects of both lives lost and years of disability generated by a conflict on a population to allow comparisons among future conflicts.

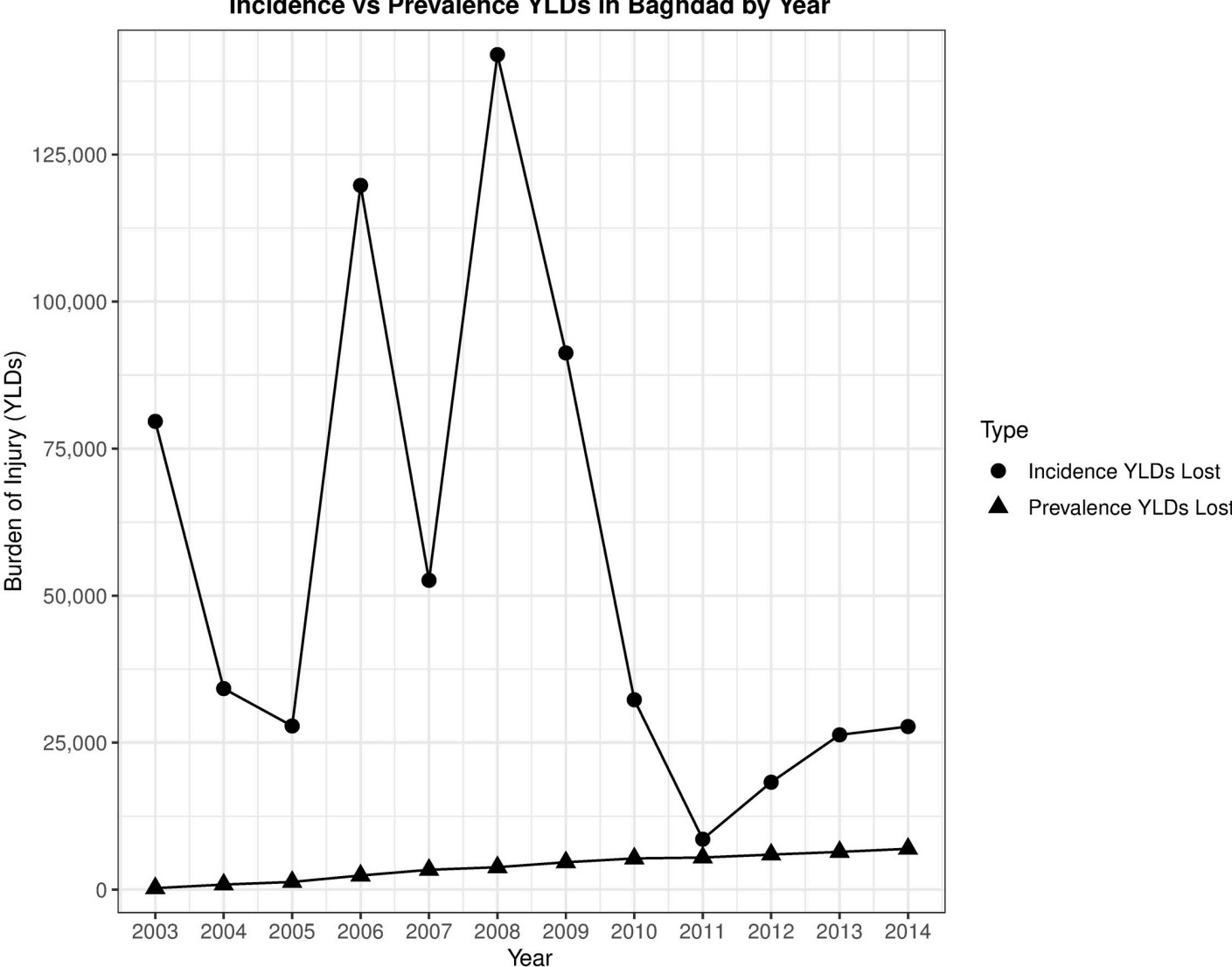

**Fig 2. Comparison of incidence-based and prevalence-based YLDs in Baghdad, Iraq, from 2003–2014.** Source of data: survey of 900 Baghdad residents in May and June 2014, regarding household injury events between 2003 and 2014. YLD, year lived with disability.

**Table 1. Proportion of DALYs by cause, total DALYs by cause for the sample, and estimated DALYs for the population of Baghdad, Iraq, with 95% UI.**

| Injuries-specific mechanism | Proportion of DALYs | Sample total DALYs | Population estimate DALYs | 95% UI (DALYs) |
|---|---|---|---|---|
| Gunshots | 56.8% | 2,348 | 3,166,000 | 2,277,000–4,175,000 |
| Shelling | 16.0% | 662 | 893,000 | 460,000–1,396,000 |
| Other | 14.4% | 596 | 803,000 | 372,000–1,276,000 |
| Blasts and explosions | 9.7% | 401 | 540,000 | 227,000–912,000 |
| Stabbing | 1.5% | 61 | 82,000 | 17–248,000 |
| Burns | 1.7% | 70 | 93,800 | 0–286,000 |

Source of data: survey of 900 Baghdad residents in May and June 2014, regarding household injury events between 2003 and 2014.

DALY, disability-adjusted life year; UI, uncertainty interval.

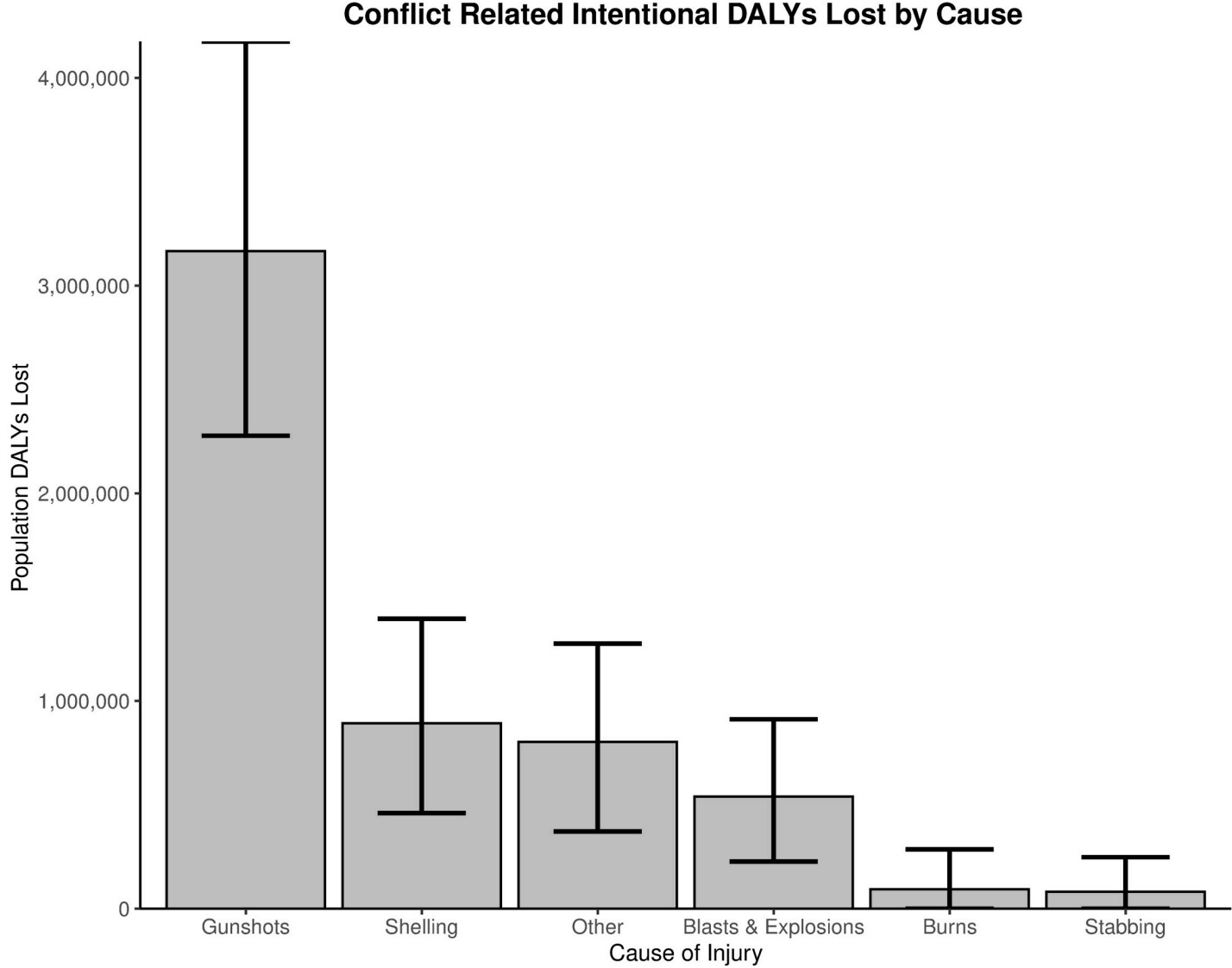

**Fig 3. DALYs lost to each identified cause of intentional injury in Baghdad, Iraq, 2003–2014. Source of data: survey of 900 Baghdad residents in May and June 2014, regarding household injury events between 2003 and 2014.** DALY, disability-adjusted life year.

Because of the, to our knowledge, novelty of this method of calculation and its application to a conflict-specific data set, it is difficult to compare our results with other settings. We did find a study from South Africa where a team used local data combined with GBD techniques to provide an estimate of DALYs lost attributable to interpersonal violence. Matzopoulos and colleagues calculated that an estimated 2,605 DALYs per 100,000 population were lost to homicide and other interpersonal violence in South Africa [28]. By our estimates, Baghdad lost 5,320 DALYs per 100,000 population in 2008 and 1,390 DALYs per 100,000 population during 2012, the highest and lowest burden of injury years in our data set. These comparisons should be made cautiously, however, as the South Africa data relied on modeling to fill in some data gaps and used a hybrid DALY approach, whereas we used an incidence-based DALY approach.

Our statistical techniques for estimating mechanisms of injury generated wide UIs, as noted in Table 1. As the analysis is applied to fewer and fewer individuals within the sample, when sampling a specific cause of injury, the likelihood that all injury patterns will be represented in each sample decreased, resulting in broadened UIs. Our confidence in the results, however, was increased with the finding that gunshot wounds comprised more than half the DALYs lost within the study, consistent with previous research on mechanisms of death in the Iraq War [7].

The goal of all conflict epidemiology is to better understand the health burden of conflict at a population level. Our study demonstrates how to apply these methods originally developed to measure the burden of diseases to the effects of conflict. In applying these methods to our previously published cause-specific estimates in Baghdad, we hope to capture both the cumulative burden of mortality and injury on the population, as well as demonstrate how our methods can be applied to other conflicts or populations. We believe these methods can provide a more complete picture of the burden of conflict on a population than previous methods. Our previous works have described the effect of conflict-related intentional injuries in Baghdad in terms of injuries per 1,000 years of exposure (5.0 95% CI 4.3 to 5.6) [11]. We also expressed the toll of conflict for the nation as a whole as being 4.55 deaths per 1,000 person-years (95% UI 3.74 to 5.27). While these other figures represent accepted methods for describing the health effects of conflict on a population, we developed this methodology to provide a picture of both the morbidity and mortality of a conflict with a single method and cumulative measurement.

These results demonstrate the importance of epidemiologically sound assessment of conflicts. Media reports tend to focus heavily on the more "spectacular" aspects of conflict, such as explosions, shelling, and air strikes. Passive methods of data collection, such as the Iraq Body Count, significantly underestimate the burden of injury—in this case possibly by a 3-fold undercount [9]. Both the number of injuries and total burden of injury suggest that the majority of morbidity and mortality was attributable to small arms rather than blasts or explosions. This is in line with many of the population-based surveys that identified gunshot wounds as the most common cause of death [8,10]. Even if shelling is included along with blasts and explosions, results suggest a 2-fold underestimation of the burden of conflict-related intentional injury, highlighting the importance of survey techniques such as random household surveys.

The large difference in incidence YLDs and prevalence YLDs (450 incidence YLDs versus 49.7 prevalence YLDs for the study population) is best illustrated by focusing on young individuals who sustain a lifelong disability. Using our proposed incident-based approach, a 20-year-old man who sustains a lifelong disability in 2011 (with an example DW of 0.2) would have 67 YLDs. We would allocate the entirety of his YLDs from his injury (i.e., approximately 67 years × 0.2) to a single year (2011). Using the prevalence YLD approach, 0.2 would be added to each year (i.e., 2012, 2013. . .) for the remainder of the anticipated life expectancy at the time of the person's injury. Both of these methods result in a total of approximately 13.4 YLDs; however, under an incident-based approach, all are allocated in 2011, while for a prevalence-based approach, 0.2 are allocated each year for the remainder of the individual's life.

Calculations were performed using both incidence- and prevalence-based YLDs. Prevalence YLDs are relatively stable over time and allow long-term comparison of the burden of disease among countries and are used for the current GBD studies. We concluded that incidence-based YLDs would be more helpful in understanding a time-specific conflict, because they better highlight the ebbs and flows of a conflict [24]. More specifically, we posit that the calculation of purely incidence-based DALYs provides a better picture of the long-term effects of a specific set of events on a given population. Other investigators may propose a prevalence-based approach to calculating the burden of injury for a conflict, but we believe that approach

fails to capture the speed at which modern conflicts change and evolve. For the purpose of creating a burden of disease estimate for modern, dynamic conflicts, we propose researchers use an incidence-based approach.

We were fortunate to complete the survey weeks before conditions on the ground deteriorated to the point where data collection would have been impossible. In June of 2014, the Islamic State of Iraq and Syria (ISIS) began a major offensive, resulting in the fall of Mosul and Tikrit [29].

Our participants were drawn from a population-proportionate sample of the city of Baghdad, so no adjustments were required following analysis. We did not adjust for changes in population over the course of the study, and citywide estimates were not weighted by administrative unit.

Survey participants asked to recall serious injuries in their households dating to 2003 may have failed to remember events more than a decade ago or lost members in a position to report on such events. Mock and colleagues reported on a household survey of injuries in a non-conflict area (Ghana), finding a significant rate of decline in recall after 1 month [30]. However, the same study showed that for injuries resulting in more than 30 days of disability, there was a minimal decline in recall. Any recall bias, however, only serves to make our estimates more conservative underestimates for the true burden.

Our methods present several opportunities for underestimation of the total burden of injury for this population. First, any individuals not definitively identified as having died of their injuries by the survey were calculated as YLDs rather than YLLs, and their contribution to the total would be significantly undervalued. However, by making assumptions regarding individuals who likely died of their injuries but were not definitively identified as having done so, we ran a significant risk of inflating our results in the opposite direction. Recall bias also presents a risk of undervaluing the contribution of YLDs to the total as people are less likely to forget a death than an injury. An additional aspect of recall bias is that lay reporting of certain injuries (i.e., by nonmedical professionals) might lead to certain types of injuries being remembered better than others (i.e., long bone fractures might be more memorable than soft tissue injuries). Injuries and death resulting from self-harm were also not counted as we elected to focus on injuries that directly resulted from conflict rather than its secondary effects; however, a case can certainly be made that these events should be considered in future analyses to fully capture the effects of conflict on a society. Last, we extrapolated to the population of Baghdad by multiplying the mean DALYs, YLLs, and YLDs by the population of the city. The mean numbers were achieved after bootstrapping, but this is a source for underestimation as individuals excluded from the study for lack of sufficient data to perform calculations may have increased the mean per person if included. However, it is unlikely that the inclusion of these individuals would have altered the UIs for the study. It is also possible that some households were selective in their reports to the survey teams regarding either the duration or circumstances of deaths or injuries within their households.

Public health researchers attempt assessments of the consequences of conflict in hopes of finding opportunities for primary, secondary, and tertiary prevention of population health disasters. Our contribution to conflict epidemiology goes beyond previous reports of simple crude mortality to assess long-term population health by calculating YLLs (or rendered disabled) by conflict-related intentional injuries. We include only estimates of the conflict-related intentional injuries suffered by the population of Baghdad, without consideration of unintentional injuries or downstream effects of conflict. The total burden of injury from all causes during this time frame, therefore, is certainly higher than our estimates—and it could be argued some of these, while not intentional, were otherwise conflict related.

With more sophisticated analyses, epidemiologists might better predict losses in advance of the onset of conflict, as well as assist civilian and military health systems in planning for care delivery. It could allow policy makers to better judge the health effects of various phases of the conflict in relation to military strategy. The US Army and Marine Corps Counterinsurgency Field Manual recommends, "whatever else is done, the focus must remain on gaining and maintaining the support of the population [31]." Public health efforts to better understand the effects of war on local populations would allow improved pursuit of that mission.

Findings from this study provide additional understanding of the population health effects of the Iraq War. Further study of this conflict is important given the paucity of data on the effects of urban warfare with high-velocity weapons on local populations. Future researchers may develop more sophisticated methods for addressing recall bias in long-running conflicts, as well as identify methods of accounting for the hazards of missing households who have been displaced or migrated. More sophisticated methods might also ensure the capture of all data points required for the calculation of disability-adjusted measures. When calculating DALYs, researchers may also weigh the benefits of incidence versus prevalence YLDs, based on the specific question being asked. To the best of our knowledge, ours may be the first attempt to directly assess DALYs lost in relation to injuries and disabilities caused by a war; our hope was to contribute to improved epidemiologic methods to expand the understanding of the health effects of war.

## Disclaimer

The views expressed in this paper are those of the authors and do not reflect the official policy or position of the Department of the Navy, Department of Defense, or the US Government.

## Supporting information

**S1 STROBE Guidelines Checklist. Completed Strengthening the Reporting of Observational Studies in Epidemiology (STROBE) guidelines checklist applicable to this work.**
(PDF)

**S1 Analysis Plan. Prospective analysis plan for calculation of DALYs for all injury patterns.**
(PDF)

**S1 Data Dictionary and R code Publication. Definitions of terms contained in shared data set, R code for bootstrapped calculation of incidence based DALYs, and definition of terms contained within the function.**
(PDF)

**S1 Questionnaire. English version of the survey tool designed and utilized for household survey and data collection by Lafta and colleagues during initial data collection.**
(PDF)

**S1 DALY Data. Shared data set needed for replication of this work.** The data are provided in the csv file format.
(CSV)

## Author Contributions

**Conceptualization:** Guy W. Jensen, Riyadh Lafta, Gilbert Burnham, Amy Hagopian, Abraham D. Flaxman.

**Data curation:** Guy W. Jensen, Riyadh Lafta, Gilbert Burnham.

**Formal analysis:** Guy W. Jensen, Abraham D. Flaxman.

**Funding acquisition:** Riyadh Lafta.

**Investigation:** Riyadh Lafta, Gilbert Burnham.

**Methodology:** Guy W. Jensen, Riyadh Lafta, Amy Hagopian, Noah Simon, Abraham D. Flaxman.

**Project administration:** Amy Hagopian.

**Resources:** Noah Simon.

**Supervision:** Riyadh Lafta, Gilbert Burnham, Amy Hagopian, Abraham D. Flaxman.

**Validation:** Abraham D. Flaxman.

**Visualization:** Guy W. Jensen.

**Writing – original draft:** Guy W. Jensen, Amy Hagopian.

**Writing – review & editing:** Guy W. Jensen, Riyadh Lafta, Gilbert Burnham, Amy Hagopian, Abraham D. Flaxman.

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
