## [Editor Report · Decision Letter 0]

8 Jun 2020

Dear Dr Jensen, 

Thank you for submitting your manuscript entitled "Calculating the burden of intentional injuries and disabilities in Baghdad, 2003-2014" for consideration by PLOS Medicine.

Your manuscript has now been evaluated by the PLOS Medicine editorial staff and I am writing to let you know that we would like to send your submission out for external peer review.

Kind regards,

Artur Arikainen,

Associate Editor

PLOS Medicine

---

## [Decision Letter · Decision Letter 1]

10 Jul 2020

Dear Dr. Jensen,

Thank you very much for submitting your manuscript "Calculating the burden of intentional injuries and disabilities in Baghdad, 2003-2014" (PMEDICINE-D-20-02532R1) for consideration at PLOS Medicine. 

Your paper was evaluated by a senior editor and discussed among all the editors here, as well as being evaluated by three independent reviewers, including a statistical reviewer. The reviews are appended at the bottom of this email and any accompanying reviewer attachments can be seen via the link below:

[LINK]

In light of these reviews, I am afraid that we will not be able to accept the manuscript for publication in the journal in its current form, but we would like to consider a revised version that addresses the reviewers' and editors' comments. Obviously we cannot make any decision about publication until we have seen the revised manuscript and your response, and we plan to seek re-review by one or more of the reviewers. 

We expect to receive your revised manuscript by Jul 31 2020 11:59PM. Please email us (plosmedicine@plos.org) if you have any questions or concerns.

We look forward to receiving your revised manuscript. 

Sincerely,

Emma Veitch, PhD

PLOS Medicine

On behalf of Clare Stone, PhD, Acting Chief Editor,

PLOS Medicine

plosmedicine.org

*We'd suggest revising the title according to PLOS Medicine's style, ideally this should include some indication of the study design as part of the title after a colon (eg, XXYY: "A randomized controlled trial," "A retrospective study," "A modelling study," etc.) 

*Please structure the abstract using the PLOS Medicine headings (Background, Methods and Findings, Conclusions - "Methods and Findings" is a single subsection).

*In the last sentence of the Abstract Methods and Findings section, please describe the main limitation(s) of the study's methodology.

*At this stage, we ask that you include a short, non-technical Author Summary of your research to make findings accessible to a wide audience that includes both scientists and non-scientists. The Author Summary should immediately follow the Abstract in your revised manuscript. This text is subject to editorial change and should be distinct from the scientific abstract. Please see our author guidelines for more information: https://journals.plos.org/plosmedicine/s/revising-your-manuscript#loc-author-summary

*We'd suggest revising the quantitative estimates in the abstract so these are clearer and relate more precisely to the estimates given in the main text, ie an point estimate (of central tendency) with uncertainty intervals/confidence intervals. 

*Ideally, please update the in-text citation callout format (this should be sequential numerals in square brackets - ie [1], [2] etc) - if using referencing software this should be fairly quick and easy.

*Please clarify in the paper if the analytical approach used here corresponded to one laid out in a prospective protocol or analysis plan. Please state this (either way) early in the Methods section.

Comments from the reviewers:

Reviewer #1: I confine my remarks to statistical aspects of this paper. The general approach is fine, but I have some issues to resolve before I can recommend publication.

It would be better to calculate life expectancy at time of injury or death. The authors use either a world wide figure or one specific to Iraqis, but both of those are calculated at birth. The life expectancy of an adult is longer than the life expectancy at birth minus current age. If this can't be done here, some mention should be made of why not.

p. 4 Choosing 86 years for life expectancy seems wrong. I think the Iraqi figure used in the sensitivity analysis should be used in the main analysis. (But see above).

p, 5 The two assumptions in the first paragraph both seem dubious. 

 I commend the authors for using bootstrapping and for making their code available

 Were there any differences between the 179 respondents and teh 46 who refused?

Peter Flom

Reviewer #2: Big picture thoughts:

It is stunning that outright deaths constituted the vast majority of years lost. This will be surprising to many and is important for thinking about the costs of conflict and in future eras the costs of reparations.

It is rare that a second analysis of a survey contributes something new and important, this is an exception.

There is a strong tendency for DALY analyses to come from large (e.g. DHS, MICS) datasets that were collected for some other purpose. This is a great example for a dataset collected for exactly this purpose (in this case, the burden of intentional injuries).

My one big criticism is that the discussion and conclusions read like an advertisement for DALY-type analyses but the results argue the opposite. A simple mortality survey is much easier to do than this complicated interview, asking the researchers to make difficult medical assessments (thus all interviewers were physicians) and bootstrap analyses. A simple mortality survey is closer to the cause of the problem in terms of the causal pathway. A simple mortality survey will likely have more political effect, be easier for the layperson to understand, and is closer in time and mechanism to all accountability processes like an ICC trial. If 90% of burden came directly from death, this study screams that we should focus on getting local professionals to focus on the easier more influential method. The issue is, perhaps with the Uyghurs, or the Rohingya, or other crises the Baghdad experience is not the norm. Exploring that in other settings seems to me the important value of this technique at this moment in history. 

Small thoughts:

Methods

Rare and great that the interviewers where physicians.

"Because life expectancy represents the aspirational healthy life span for all individuals, we followed the GBD methodology of selecting the lowest observed death rate for any age group in countries of more than 5 million in population, or 86 years.22" While this might be referenced, it is so profound that it beckons a line or two more explaining what the reference in this case was (e.g. based on 140 countries across the globe or based on the 11 countries in the Middle-east over 5 million population or based on Baghdad in 2000 with some projection model?). Moreover there is a danger that future people will repeat this process in a completely different setting and the readers will not understand that difference will be because of the assumed life expectancy. Thus, it would be ideal if in the methods there could be a table of the male and female life expectancies at birth, 10, 20, 30, 40, 50, 60 ,70, 80, ...>85=0 or something like that. 

YLLs have an outsized effect on total DALY calculations, therefore conservative practice required YLLs to be calculated only for individuals who definitively died of their injury." The implications of this are not covered in the discussion and should be. 

Results

You say, "Forty-six individuals with intentional injury could not be used in the calculation of DALYs for the sample due to missing data points." So then is the denominator you used to assessing the burden of disease 5148 individuals, or some fraction of that population corrected for the incomplete data (e.g. 175/229 X 5148 would be one extreme while using 5148 as the denominator would be the other extreme)? Just ignoring those 46 with incomplete data would be markedly under-estimating the burden of injury and might be explaining why outright deaths are the lion's share of DALY's lost. Perhaps you did a sensitivity analysis (e.g. exclude vs. assume they had lost the average # DALY's as those with complete records....), but if so, I somehow missed it.

Discussion

"Both the number of injuries and total burden of injury suggest the majority of morbidity and mortality was attributable to small arms rather than blasts or explosions." Might be worth mentioning that the western press (as reflected by IBC and others) suggested explosions accounted for most violent deaths. This highlights the advantages of random HH surveys like this.

There are several mechanisms by which this is a likely under-estimate (if cause of death was blurry, excluding injuries where the data was not complete....) and this is not adequately addressed.

Reviewer #3: I read the revised version of the manuscript titled as "Calculating the burden of intentional injuries and disabilities in Baghdad, 2003-2014". The subject is important and interesting, but I think there are some issues in the manuscript. 

Abstract

- "YLLs were calculated by subtracting the age at death from the age the subject might have expected to achieve at birth": This is not correct, and is not consistent with the Methods section in the full text. For calculating YLL, instead of life expectancy at birth, life expectancy at the age of death is used. 

- I think "Finding" section of the abstract should include a little bit more of the important findings of the manuscript. 

Methods

- Self-harm is not included in the study, so they cannot be considered equal to intentional injuries. It seems that the study covers include conflict and terrorism, as well as interpersonal violence. I think the whole manuscript should be checked for consistency.

- To calculate YLD, both external causes and nature of injuries should be available. It is not clear how a population survey is valid to extract frequency of different types of bodily injuries (nature of injury); non-medical professionals might remember injuries such as fracture in large bones, but usually are not capable to differentiate for instance a vascular injury from other soft tissue injuries. Also, there is a high probability of forgetting separate minor injuries or minor injuries in the context of a multiple trauma. Also, it is not clear how a cross-sectional survey can be used as a valid tool to estimate both prevalence and incidence during a relatively long period of time. I think it is necessary to explain a little more around the survey and validity of its data that have been used as the main source for this study. 

- It is not clear how the number of fatalities has been estimated through such survey.

- The data are related to a specific period of time and cannot be updated (not sure why not published earlier), but is there any specific reason for authors preference on using GBD 2013 methodology (while GBD methods are continuously updated)

Findings

- It is better to provide some data on total number of deaths and injuries, before providing estimates for YLL, YLD and DALYs. 

Discussion

- Is there any reason for the relatively huge difference between incidence-based and prevalence-based YLDs? Considering the fact that duration of disability is considered in calculations?

- There is no comparison with other studies; the estimated at least could be compared with the most recent GBD estimates for the same cause and same period of time

[LINK]

---

## [Decision Letter · Decision Letter 2]

5 Nov 2020

Dear Dr. Jensen,

Thank you very much for submitting your manuscript 'Calculating the burden of conflict-related intentional injuries and disabilities in Baghdad, 2003-2014: A modeling study and proposed method for calculating burden of injury in conflict' (PMEDICINE-D-20-02532R2) for consideration at PLOS Medicine. 

Your article was also evaluated by our editorial team and one of the original independent reviewers from the earlier review round. Additionally, we received comments from a new reviewer, as one of the reviewers from the first round was unavailable to review the revised text. Please find their comments are pasted below.

[LINK]

We would like to consider a revised version that addresses the reviewers' and editors' comments. Please note that we cannot make any decision about publication until we have seen the revised manuscript and your response, and we may seek re-review by one or more of the reviewers. 

In revising the manuscript for further consideration here, your revisions should address the specific points made by each reviewer and the editors. Please also check the guidelines for revised papers at http://journals.plos.org/plosmedicine/s/revising-your-manuscript for any that apply to your paper. In your rebuttal letter you should indicate your response to any comments from reviewers or editors and the changes you have made in the manuscript. Please submit a clean version of the paper as the main article file; a version with changes marked should be uploaded as a marked up manuscript. 

In addition, we request that you upload any figures associated with your paper as individual TIF or EPS files with 300dpi resolution at resubmission; please read our figure guidelines for more information on our requirements: http://journals.plos.org/plosmedicine/s/figures.

While revising your submission, please upload your figure files to the PACE digital diagnostic tool, https://pace.apexcovantage.com/ PACE helps ensure that figures meet PLOS requirements. To use PACE, you must first register as a user. Then, login and navigate to the UPLOAD tab, where you will find detailed instructions on how to use the tool. If you encounter any issues or have any questions when using PACE, please email us at PLOSMedicine@plos.org.

We expect to receive your revised manuscript by . Please email us (plosmedicine@plos.org) if you have any questions or concerns.

Your article can be found in the 'Submissions Needing Revision' folder. 

We look forward to receiving your revised manuscript. 

Sincerely,

Artur Arikainen, 

Associate Editor 

PLOS Medicine

plosmedicine.org

1. Please address the reviewers’ comments below.

2. Please place the Author Summary after the Abstract.

3. Author Summary:

a. Please use the 3 standard PLOS Medicine subheadings listed here: https://journals.plos.org/plosmedicine/s/revising-your-manuscript#loc-author-summary. 

b. Please use bullet points under the 3 subheadings.

c. Lines 43-44: Please rephrase this for clarity to a lay reader: “Previous researchers innovated the population-proportionate to size sampling methodology for household surveys in war zones…”

4. Abstract: 

a. Please include summary participant/household demographics.

b. Line 87: Please define DALYs at first use.

5. Please move citations before punctuation, and remove spaces within the square brackets, eg: “…increasingly sophisticated [1,2].”

6. Line 293 Please mention if consent was “informed”.

7. Line 370: Please rename this section “Discussion”. Please present and organize the Discussion as follows: a short, clear summary of the article's findings; what the study adds to existing research and where and why the results may differ from previous research; strengths and limitations of the study; implications and next steps for research, clinical practice, and/or public policy; one-paragraph conclusion.

8. Please include the study questionnaires as Supporting Information files.

9. Please rename the analysis plan file ‘S1 Text’ and update the reference at line 197.

---

Comments from the reviewers:

Reviewer #1: The authors have addressed my concerns and I now recommend publication

Peter Flom

-

Reviewer #4: The paper focuses on measuring the impact of civil war using mortality as an indicator of severity. It proposes an alternative to previously used demographic approaches which calculated mortality rates based on data from household surveys. Here the authors still use household surveys but express the results in years of life lost or years lived with disability. They estimate about 4 - 6 million life years lost and 400 000 - 900 000 years lived with disability. Disability as the authors point out, is a persistent and socially significant problem especially among communities who have gone through a major conflict. The issue is also neglected and hence the paper is a welcome addition to the literature.

Previous reviews have raised detailed questions on the statistical aspects of the paper and the authors appear to have addressed them. I find two main weaknesses, which if handled, would strengthen this paper substantially.

First, the selected sampling units were extrapolated to whole of Baghdad. (line 281) This requires better explanation of how these units were selected and why the authors consider them representative of the city and therefore plausibly extrapolated to the whole city of about 7 million. Civil conflict, as is well known is an unpredictable affair in the ways it the action distributes itself. Areas of cities may remain untouched (e.g. Aleppo in the Syrian war) and some areas completely destroyed. The authors may have tested the sample for biases (a 5.5 household size in sample against 6.5 household size from the World Bank) but they do not present their work in the paper.

The section on Methods and findings begins with data collection without even a cursory description of the sampling design, household selection, construction of the sampling frame - all of which are foundational for what is to come - that is extrapolating the results from the 900 household. Why these 900 households are representative of the entire city population (about 7 million) is not explained despite being a critical pillar for the credibility of the final estimations (e.g. the sample in the study seems to have a family size of 5.5 although the family size I believe it's about 6.5 according to the World Bank. The section describes perhaps in disproportionate detail the YLL and YLD methods at the cost of explaining the sampling. The YLL and YLD methods are well known and could have been cited to the GBD documents. On the other hand, the sampling, its biases and the household selection underpins the very crux of this paper - extrapolation of the sample to the entire city and hence the final results. 

Moreover the study applies standard YLL and YLD methods to data and that, per se, is not innovative. What maybe innovative is that this well-developed method is applied to Baghdad which had gone through a war. It may be a better idea to emphasize that aspect more than an innovation on the method. 

Secondly, the results are a bit indecipherable since they don't make reference (Line 381 - 384) to anything else for the reader to compare - even YLL and YLDs made by GBD in poor countries without a conflict or indeed cities with an lot of urban violence such as Medellin or Lagos. What does 5.6 million DALYs mean in relation to other settings? It seems an huge number but is it? Or is this implausible. Some clarity on this would make the results much more relevant and understandable. 

I am sure the authors are fully knowledgeable of all of these issues that I have described. Addressing them would strengthen the paper substantially.

---

[LINK]

---

## [Editor Report · Decision Letter 3]

7 May 2021

Dear Dr. Jensen,

Thank you very much for re-submitting your manuscript "Calculating the burden of conflict-related intentional injuries and disabilities in Baghdad, 2003-2014: A modeling study and proposed method for calculating burden of injury in conflict" (PMEDICINE-D-20-02532R3) for review by PLOS Medicine.

I have discussed the paper with my colleagues and the academic editor. I am pleased to say that provided the remaining editorial and production issues are dealt with we are planning to accept the paper for publication in the journal.

[LINK]

We look forward to receiving the revised manuscript by May 14 2021 11:59PM.   

Sincerely,

Louise Gaynor-Brook, MBBS PhD

Associate Editor 

PLOS Medicine

plosmedicine.org

Comments from the Academic Editor:

The only additional suggestion to reinforce is one of the reviewers' requests to put these numbers (particularly injuries) in some context with previous work--is is a lot or not so many? The authors replied that this is difficult to do since their methods are different from past literature, however I think it would still be helpful to cite findings from from conflict situations that are broadly similar for context.

Requests from Editors:

General comments: Please try to address the comments by reviewers and the Academic Editor that some context/comparison for your findings would be helpful, in your revised text. 

Please be consistent in the use of hyphenation of ‘conflict-related’ throughout the text 

You have recorded in the submission form questionnaire that ‘My study does not require an ethics statement.’ If possible, please correct this to reflect the details of the ethical approval detailed in your methods section. 

Title: Please revise your title according to PLOS Medicine's style. We suggest “Conflict-related intentional injuries and disabilities in Baghdad, Iraq, 2003-2014: A modeling study and proposed method for calculating burden of injury in conflict” 

Abstract Background: Please expand upon why the study is important; what are the issues with using mortality as the primary measure of the population health effects of war? 

Abstract Methods and Findings: 

Please ensure to add that this is a modelling study. 

Please provide brief demographic details on the population interviewed; gender, age, etc. 

Please provide more detail on the data collected during interviews e.g. self-reported injury, ability to work, etc. which has contributed to your conclusions 

Please move the paragraph beginning “”Our data come from interviews…” to the beginning of the Abstract Methods and Findings

Line 45 - Please consider rephrasing to “YLLs were calculated by subtracting the age at death from the expected age of death”. Please clarify what the expected age of death is based on.

Line 54 - please define UI at first use. Please revise 4.9 million to 4.99 million (as shown on line 59) if this is correct. 

Line 59 - there is no need to repeat results; please omit “4.99 million YLLs lost (95% UI 3.87 million – 6.13 million) versus 616,000 YLDs lost (95% UI 60 399,000 – 894,000).”

Please begin your Abstract Conclusions with “"In this study, we calculated ..." or similar, to summarise your findings for the reader.

Author summary

Within ‘What Do These Findings Mean?’,

Please provide a non-technical summary of your results 

Please include a final sentence outlining the limitations of your study 

Please consider re-ordering the first two bullets points 

Please rename ‘Background’ as ‘Introduction’ 

Line 115 - please omit sentence beginning ‘This paper reports a new approach…’

Methods

Please separate your ‘Methods and Findings’ section into separate ‘Methods’ and ‘Results’ sections

Please ensure that the study is reported according to the STROBE guideline, and include the completed STROBE checklist as Supporting Information. Please add the following statement, or similar, to the Methods: "This study is reported as per the Strengthening the Reporting of Observational Studies in Epidemiology (STROBE) guideline (S1 Checklist)." The STROBE guideline can be found here: http://www.equator-network.org/reporting-guidelines/strobe/ When completing the checklist, please use section and paragraph numbers, rather than page numbers.

Please ensure that where methods are described elsewhere e.g. references 11 and 19, that the papers cited are open access. If not, please ensure to describe the methodology more thoroughly. 

Please confirm whether the analysis plan was prospective 

Line 237 - please omit ‘per Salomon’ (citation will suffice). 

Line 314 - please quote the actual number/percentage of individuals excluded due to missing data in place of unspecific terms such as ‘nearly’

Discussion 

Please remove subsection headings within the Discussion (e.g. Total DALYs Lost, and so on)

Please begin your Discussion with a short, clear summary of the article's findings.

Line 390 - please take note of the comments from the Academic Editor “to put these numbers (particularly injuries) in some context with previous work--is is a lot or not so many? [Although] methods are different from past literature… it would still be helpful to cite findings from from conflict situations that are broadly similar for context”. 

Stylistically speaking, please add ‘to the best of our knowledge’ or similar to avoid assertions of primacy (e.g. "Because of the novelty of this research", “ours may be the first attempt…”)

References

Please ensure that references are appropriately formatted and journal titles are appropriately capitalised e.g. Annual Review of Public Health. Journal name abbreviations should be those found in the National Center for Biotechnology Information (NCBI) databases. Please see https://journals.plos.org/plosmedicine/s/submission-guidelines#loc-references for further guidance.

Figures

Figure 1 - please correct DALYs on the y axis 

Supplementary files: 

Line 302 - “We have provided the code and data set in our online appendix. “ Please ensure that either the relevant file is appended to your resubmission, or that the freely / publicly available location of the data is shared in your manuscript. The file ‘Data Dictionary and R Code Publication-adf-ns.docx.pdf’ appears to be an unrevised version of your manuscript.

[LINK]

---

## [Editor Report · Decision Letter 4]

25 May 2021

Dear Dr Jensen, 

On behalf of my colleagues and the Academic Editor, Prof. Margaret Kruk, I am pleased to inform you that we have agreed to publish your manuscript "Conflict-related intentional injuries in Baghdad, Iraq 2003-2014: A modeling study and proposed method for calculating burden of injury in conflict" (PMEDICINE-D-20-02532R4) in PLOS Medicine.

PRESS

Sincerely, 

Louise Gaynor-Brook, MBBS PhD 

Associate Editor 

PLOS Medicine